# Discerning the Effects of Rural to Urban Migrants on Burglaries in ZG City with Structural Equation Modeling

**Fangye Du** [1,2]**, Lin Liu** [3,4],*[ID]**, Chao Jiang** [5],*[ID]**, Dongping Long** [3] **and Minxuan Lan** [4][ID]

1 Key Laboratory of Regional Sustainable Development Modelling, Institute of Geographic Sciences and Natural Resources Research, CAS, Beijing 100101, China; dufy.18b@igsnrr.ac.cn
2 College of Resources and Environment, University of Chinese Academy of Sciences, Beijing 100049, China
3 Center of Geographic Information Analysis for Public Security, School of Geographic Sciences, Guangzhou University, Guangzhou 510006, China; longdongping2012@163.com
4 Department of Geography, University of Cincinnati, Cincinnati, OH 45221-0131, USA; lanmn@mail.uc.edu
5 Institute of Remote Sensing and Geographic Information Systems, School of Earth and Space Sciences, Peking University, Beijing 100871, China
* Correspondence: lin.liu@uc.edu (L.L.); zsjc_2007@126.com (C.J.)

**Abstract:** Both rural to urban migration and urban crime are well researched topics in China. But few studies have attempted to explore the possible relationships between the two. Using calls for service data of ZG city in 2014, the Sixth Census data in 2010, this study examines relationships between migrants and crime by using structural equation models. Two hypotheses were tested: (1) the distribution of migrants has direct effects on the spatial distribution of burglaries, and (2) migrants also indirectly affect burglary rate through mediating variables such as residential mobility and socio-economic disadvantage of their resident communities. The results showed that migrants have significant direct and indirect effects contributing to burglaries, although the indirect effect is much larger than the direct effect, indicating that community characteristics play a more important role than the migrants themselves.

**Keywords:** migrant; burglary; structural equation model; PLS; China

---

## 1. Introduction

The research on the relationship between immigration and crime has gained much attention in crime research due to its significance for policy implication. Empirical studies of the impact of immigration on crime in western countries [1,2] have shown inconsistent findings. Some researchers identified that immigration may lead to higher rates of crime, which corroborate traditional theories such as social disorganization theory [3,4], while others revealed that immigration has no or little impact on crime or even reduces crime rates [1,5]. The contradictory conclusions from these studies suggest that how immigration affects crime may vary with the specific social environment. In the context of China, immigrants from foreign countries are very scarce compared to western countries; instead, domestic migrants from rural areas to cities are very significant, with the total number exceeding 274 million people for China in 2014.

China's household registration system (HRS), or Hukou, has long separated rural residents from urban residents, with rural areas focusing on agriculture while urban areas specialize in industry and business. This system poses constraints on the free movement of the population between urban and rural areas [6]. Although the original intent of HRS was to achieve shared prosperity across the nation, rural residents have long been placed in a disadvantaged position due to uneven allocation of

resources favoring large cities. The great gap between urban and rural regions has become a serious societal issue in China. After the reform and opening policy was implemented in China in 1978, many migrants started to flow into the cities to seek employment opportunities [7]. However, the migrants, who do not have the city Hukou, are not eligible to all the benefits and services that residents are entitled to. This contributes to a series of social issues, among which crime has become one of the most serious ones that has a great impact on urban safety.

Curran and Denial (1998) analyzed city-based migrants and crime rates, using mostly descriptive approaches. Aside from the work of Liu et al. [8], the relationship between migrants and crime in China has received little attention. To this day, no research has been conducted to analyze the direct and indirect relationships between migrants and crime in China.

In this study, we choose burglary as our focus as it is one of the major crime types and is closely related to the community environment. From a spatial perspective, this paper examines the direct and indirect effects of migrants on the spatial pattern of burglaries in the city of ZG, China. With the Sixth Census data and 2014 burglary data for 1993 communities of ZG, a structural equation model (SEM) is constructed. Two hypotheses are tested: (1) the distribution of migrants have direct effects on the spatial distribution of burglaries, and (2) migrants also indirectly affect burglary rates through mediating variables such as residential mobility and socioeconomic disadvantages of their residing communities. Furthermore, this study uses the context-specific variables as controls to ensure the validity of these effects.

## 2. Literature Review

This section reviews the research conducted in western countries about the relationship between immigrants and crime, followed by the limited research on the relationship between migrants and crime in China.

### 2.1. Crime-Generating Effect of Immigrants

The impacts of immigrants on crime have been found to be complex as conflicting results were obtained from different empirical studies [4,5,9–11]. On one hand, some traditional perceptions argue that immigrants have positive associations with crime. Three major theoretical perspectives have been developed or used to explain why immigrants contribute to crime, including opportunity structure, cultural approaches, and social disorganization [12–16].

The opportunity structure view argues that different groups of the community do not have equal access to opportunities. For example, migrants, as the disadvantaged group, cannot register their children in the local public schools to receive free education. This perspective is consistent with the strain theory proposed by Merton (1938) [12], which suggests that legitimate opportunities for wealth and social status are not equally available to all groups. Both notions mention that immigrants characterized by low levels of skill, education, and income typically have fewer legitimate opportunities and are not eligible for all social welfare. Since they experience a greater level of financial difficulty, they are more likely to take advantage of illegal opportunities, e.g., stealing, to overcome blocked economic and employment opportunities [17].

Besides the opportunity structure in a community, the cultural approaches notion explains the crime-generating effect of immigrants from the perspective of human behavior. In line with cultural deviance theory [13], this framework argues that criminal behavior can be learned among each other and crime would become a normal response to the opportunity structure of a community. On the other hand, when cultural conflicts exist between immigrants and native residents, the social integration of native and non-native residents can be low, which disturbs collective efficacy of the community by weakening informal guardianship [18].

Social disorganization theory [14] has been widely revisited by many researchers and they contend that crime is more likely to occur in a "social disorganized" community marked by high levels of socio-economic disadvantage (SED), ethnic heterogeneity, and residential mobility [5,16,19,20].

The research on immigration and community has pointed out that a potential impact of immigrants on a community is to disorganize and destabilize its social structure [21]. As such, the community with a high proportion of immigrants is expected to experience more crime activities since the residential stability is weakened.

A large body of empirical research has been conducted in different areas with various methods to support these theoretical perspectives [4,11]. For example, Borjas and Grogger (2009) found that a 10% increase in African-American immigrations in the U.S. is associated with about a 1.2% increase in incarceration rate during 1960 and 2000 [22]. They explained that the additional labor force competed for the limited employment opportunities and lowered wages for the job market, making people more likely to pursue illegal opportunities. Using the decennial census data from 1980–2000 and least square estimation, another empirical study conducted in the U.S. demonstrated that immigrants from Mexico increased property crime by disturbing economic conditions, labor market outcomes, and quality of housing in counties [4]. Additionally, some research indicated that immigrants with diverse cultural backgrounds and limited economic resources could weaken community institutions [23]. The loss of community institutions consequently led to increased criminal offending for both immigrants and native-born residents in such communities [19].

### 2.2. Crime-Decreasing Effect of Immigrations

In contrast with the aforementioned studies, another group of research has provided the opposite view, arguing that immigration has no effect on crime, or even lessens it [24]. This is more in line with the theoretical perspective of emerging community resources. This perspective argues that immigrants can promote the economy in the destination community because they not only provide low-waged labor for employers for high profit but also attract new resources for the community such as employment opportunities, investments, and infrastructures [25].

There are empirical studies to support this crime-decreasing effect of immigration. For example, individual immigrants are found to be less likely to commit crimes than their native-born counterparts in the U.S. [5,17]. The effects of immigrants on the social organization of a community are found to be minimal in both U.S. and western European countries [26]. Similarly, Wadsworth et al. (2010) offered insights into the complex relationships between immigrants and crime, and suggested that growth in immigration may facilitate the precipitous crime drop during the 1990s in the U.S. [27]. From a cultural perspective, Ignatans and Roebuck (2018) found that the areas with the most European and African immigrants have the lowest average crime rates in England and Wales [28]. Their results also suggest that the cultural similarity between the migrant and indigenous population is a key determinant of whether immigrants increase or decrease crime.

Several methods have been used in these empirical studies such as dual multivariate latent regression, multiple linear regression, fixed effects regression, and the structural equation model [2,26]. For example, Reid (2005) constructed a multiple linear regression model to examine the effects of immigration on crime in the metropolitan areas of the U.S. and they identified negative association between immigrants and crime [1]. Using the fixed effects regression model, Ferraro (2016) addressed the effect of immigration on overall, violent, and property crime in U.S. cities that have experienced significant growths in immigration during 2000–2007 [24]. Their results also supported the significant impact of immigrants on reducing crime. Further, Hagan and Palloni (1999) developed a logit model to compare the involvement of Hispanic immigrants and U.S. citizens in crime and they found that Hispanic immigrants were less engaged in criminal activities because the immigrants were found to do as well as or even better than the U.S. citizens [29].

In summary, the impacts of immigrants on crime are complex, involving both crime-generating and crime-decreasing effects, which also implies that the relationship between immigrants and crime is contingent upon specific social contexts and environments. In the context of China, the outsiders to a community are dominated by the migrants who come from rural areas and how these migrant populations affect crime has not been fully discovered yet.

### 2.3. Migration and Crime in China

As mentioned earlier, HRS creates a deep divide between urban and rural populations and many migrants flow into the cities to look for job opportunities. Since they are the marginalized group of the population, they face many challenges which may lead to a series of social issues [8]. For example, Wang (1999) identified that migrants in Shanghai faced the problems of unemployment, temporary housing, residential segregation, SED, and institutional barriers [30]. Based on a survey conducted in 2005, Lu (2013) also found that migrants in Shanghai were exposed to the inequality of economic and opportunities with OLS (Ordinary Least Squares) regression [31]. Due to the low socio-economic status (SES) and the fewer opportunities of the migrants, they inevitably increase SED and residential mobility in the community.

As a result, empirical studies conducted in China about the impacts of migrants on society largely support the three theoretical perspectives of opportunity structure, cultural approaches, and social disorganization. For example, Hu (1998) discussed the potential social issues brought by a large number of migrants [32]. With the arrest data in Guangzhou, he showed that migrants engaged in more criminal activities than the registered residential counterparts, and concluded that the concentration of migrants contributed to the high crime rate. At the community level, using migrants' data and crime data from China Statistical Yearbook during 1978–2003, Lo and Jiang (2006) found that migrants are more likely to be involved in criminal activities because they are less bonded to the community and challenged by social inequality and limited opportunities [33]. Using Kernel density estimation with longitudinal data from 2009 to 2012 in Tong County, Zhejiang Province, Jin and Li's research (2014) drew the conclusion that migrants and crime incidents were largely co-located in space [34]. In an examination of the relationship between neighborhood environment and residential locations of juvenile and adult migrant burglars in China, Liu et al. (2018) revealed that a neighborhood with high residential instability is likely to attract more juvenile migrant burglars, while a socially disorganized neighborhood tends to include more adult migrant burglars [8]. Limitations, such as the lack of explicitly considering the effect of culturally different migrants who live in the same community [28], do exist for these three theories. However, the cultural issue in China is not as pronounced as that in Europe or America, therefore we consider the immigrant as a whole in this study.

Though the effects of migrants on crime have been examined in China, empirical evidence is still limited. Particularly, the types of crime were not differentiated in many of the existing studies, meaning the unique impacts of migrants on different types of crime were ignored. Drawing upon the prior research on immigrants and crime, this study focuses on the relationship between migrants and burglaries, in an effort to discern the direct effects of rural to urban migrants on burglaries as well as the indirect effect of migrants through the mediation of resident community characteristics.

## 3. Conceptual Framework

The SEM approach is adopted in this study because it can simultaneously model complex relationships between exogenous and endogenous variables. For example, Feldmeyer (2009) constructed a SEM to examine the effects of Latino immigration on Latino homicide rate, robbery rate, and violent index rate in the U.S. [25]. Besides the hypothesized direct impact of Latino immigration on the intended crime rates, the author specifically tested the indirect effects through two latent variables including social disorganization and community resources, combined with the control built environment variables such as population density [35–38]. Drawing upon this structural framework and previous studies, we develop a conceptual framework to analyze the effects of migrants on burglaries in ZG, as shown in Figure 1.

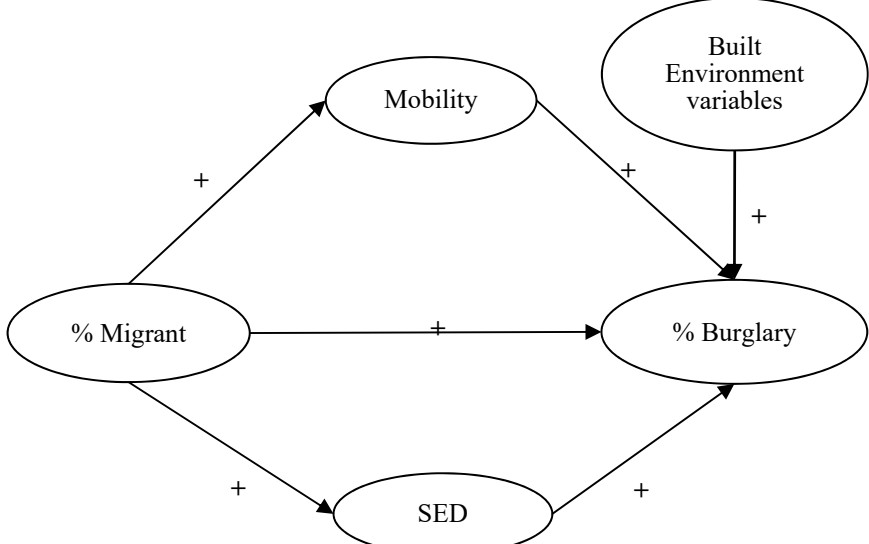

**Figure 1.** Theoretical assumptions for the direct and indirect effects of migrants on burglary rates.

First, according to past empirical evidence, we hypothesize that migrants would have a direct positive impact on burglary rates in ZG because the preliminary analysis indicated the migrants often lived in crime-prone areas, which will be further illustrated by Figure 2 in Section 4.

Second, based on social disorganization theory, we test how the mediation of community characteristics would affect the impact of migrants on burglaries. This means that migrants can influence burglaries in indirect manners. Past studies have shown that SES, residential mobility, racial heterogeneity, and family disruptiveness are important mediating factors for burglaries. For example, using spatially lagged negative binomial regression models, Nobles (2016) demonstrated that community characteristics such as residential mobility, racial heterogeneity, and family disruptiveness have a positive effect on the distribution of burglaries in Jacksonville, Florida [10].

Based on previous studies and the special context of China, we evaluated these indirect effects of migrants on crime through two mediating variables: residential mobility and SED. Specifically, we expect that the high residential mobility and low social-economic status would reduce community efficacy, which can lead to a lack of informal control over the potential offenders and insufficient guardianship over the potential targets within the community, thus generating much more opportunities for burglaries [19,20].

In addition, two spatial variables, the number of supermarkets and the number of catering facilities, are considered as control variables that affect burglary rates. Past studies have shown that a burglar is more likely to choose his/her target near urban facilities like supermarkets and catering facilities because these places usually act as crime generators [38,39]. By providing food and entertainment, supermarket and catering facilities are key routine activity nodes for both citizens and potential burglars, and thus constrain the spatial range of their movement [40,41]. As a result, the potential burglars will be more familiar with the residential areas in the vicinity than those far away. Therefore, we expect higher burglary rates to occur at the places with a high average number of supermarkets and catering facilities.

## 4. Study Area, Variables, and Method

### 4.1. Study Area

The study area is the city of ZG, a major city located in the southeast region of China, which has a total area of 7434 km$^2$ and a population of 131 million. ZG consists of 10 districts and 1993 *Juwei*s. *Juwei* is the smallest official statistics management unit whose size is about 0.5–3 km$^2$. As such, a *Juwei*

can be viewed as a community in this Chinese context. The growth rate of migrant populations in ZG is about 5.2–11% during the past 5 years. The Sixth China Census data collected in 2010 is used to extract necessary variables. The number of supermarkets and catering facilities are obtained from a navigation map provided by a local mapping company.

　　The burglary data are drawn from the call for services data of ZG in 2014, which includes detailed information on the crime type and incident address. The incident addresses are geocoded and aggregated into the 1993 communities. Among all of the calls for services, there are about 40,000 burglary incidents, accounting for about 9.42% of all calls. Figure 2 shows a zoom-in view of the central city. The yellow surface shows the density distribution of burglaries with a darker yellow representing the higher concentration of burglaries. The dots represent the community centroids and the color indicates the density distribution of migrants: red dots indicate high migrant density; yellow dots indicate medium migrant density, and green dots indicate low migrant density. The overlay of burglaries and migrants provide visual evidence to the hypothesis that migrants have a positive relationship with burglary rates. Further, the communities that have higher burglary densities are in the old town and the peripheries of ZG. These areas feature a high density of housing units, large mixed populations, a large number of facilities related with the crowd gathered and low community security, which supports the hypothesis on the indirect impacts of migrants on burglary rates.

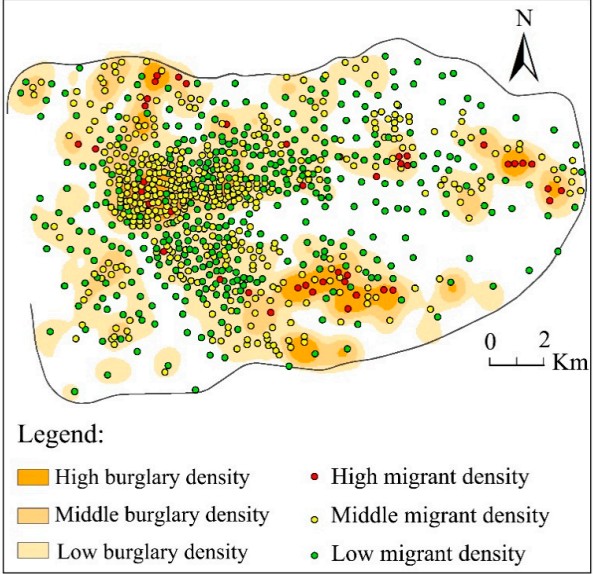

**Figure 2.** The density of burglaries and migrants in ZG (zoom-in).

## 4.2. Variable Operationalization

　　The operationalization of the latent variables is based on the framework described in Figure 1 of Section 3. Table 1 summarizes the variables used in this study and their descriptive statistics. The burglary rates are expressed as the number of burglary cases per 1000 families in the community. The key explanatory variable in this analysis is the percentage of migrants calculated by the share of migrants who do not have ZG Hukou.

　　The choice of the variable measuring mobility is made by the fact that the high proportion of renters in the community has the negative consequences of social organization, weakening formal and informal networks and limiting the action of collective efficacy, which in turn account for increasing burglary rates [3,20]. Additionally, these subgroups of the population typically lack a close bond to a residential area; thus, it would be easier for them to change their residential places compared to those who own a house or have families in the community. In the context of China and based on the data availability, we select the percentage of renters in the community as the measure of residential mobility.

**Table 1.** Descriptive statistics of structural equation model variables in the community.

| Latent Variable | Observed Variable | Min | Max | Mean | S.D. | Skewness | Kurtosis |
|---|---|---|---|---|---|---|---|
| Burglary Rates (Burglary) | The amount of burglaries per 1000 families in the community (% Burglary) | 0.00 | 14.12 | 0.97 | 1.27 | 3.77 | 24.55 |
| Migrants | The percentage of migrants in the entire population (% Migrant) | 0.00 | 0.97 | 0.32 | 0.05 | 0.74 | −0.46 |
| Residential Mobility (Mobility) | The proportion of renters in the community (% Renter) | 0.00 | 1.00 | 0.31 | 0.07 | 0.76 | −0.47 |
| Socio-Economic Disadvantage (SED) | The percentage of families with low rental costs (% LowRent) | 0.00 | 1.00 | 0.56 | 0.14 | −0.34 | −1.42 |
| | The proportion of the population who are factory workers (% Worker) | 0.00 | 0.95 | 0.27 | 0.04 | 0.88 | −0.25 |
| | The proportion of the population whose degrees are below high school (% LowEdu) | 0.69 | 0.99 | 0.76 | 0.03 | −1.33 | 1.60 |
| Control Variables | The amount of catering facilities per 1000 families in the community (% Catering) | 0.00 | 79.71 | 3.55 | 24.79 | 4.60 | 42.78 |
| | The amount of supermarkets per 1000 families in the community (% Supermarket) | 0.00 | 26.55 | 1.72 | 6.58 | 3.43 | 18.32 |

A hallmark of a disorganization area is SED [14]. The communities faced with SED are prone to generate more crime opportunities. Based on the context of China and data availability, SED was measured by the percentage of families with low rent cost, the proportion of the population who are factory workers, and the proportion of population who do not have a high school degree.

Additionally, we select two spatial factors as control variables: the average number of supermarkets and the number of catering facilities [38,40]. The control variables allow the effect of other variables to be accounted for when evaluating the impacts of migrants on burglary rates.

The potential collinearity problem is checked through the variance inflation factor (VIF) and no significant correlation is found among the variables. In addition, correlations between burglary rates and explanatory variables are also checked with Spearman's correlation rank and the results indicate that all explanatory variables are significantly correlated with burglary rates ($p < 0.05$).

### 4.3. Method

SEM, which integrates variance analysis, regression analysis, path analysis, and factor analysis, is a complex multidimensional relationship modeling approach [42]. SEM normally consists of two components: the measurement model and the structural model. The measurement model specifies the relations between a latent variable and its observed indicators, whereas the structural model specifies exogenous and endogenous variables as well as the relationships between them [43]. With SEM, we can estimate to what extent exogenous latent variables are able to predict endogenous latent variables [44]. Especially, the total effects of an exogenous variable on an endogenous variable are obtained by combining both direct and indirect effects. The existence of indirect effects means that the exogenous variable can influence the endogenous variable through mediating variables. Since this study attempts to examine the complex effects of migrants on burglary rates, some of which are realized through mediating factors such as SED.

There are six steps involving building a structural equation model, including model specification, implied covariance matrix determination, model identification, model estimation, model evaluation, and model re-specification [42]. It is important to note that model specification needs to be established based on the theories and the previous literature. In this study, the model is specified based on the conceptual framework explained earlier (Figure 1). Since part of variables do not follow the normal distribution as indicated by the skewness in Table 1, partial least squares regression estimate (PLS) is used [45]. The model evaluation includes two components. First, the adjusted $R^2$ is used to evaluate the fit of the structural model. Second, the fit of the measurement model can be assessed by the composite reliability, Cronbach's coefficient alpha (CCA), and average variance. Composite reliability (CR) is an estimate of the reliability of construct scores. Higher CR indicates higher consistency among variables. CCA evaluates the reliability of the measurement model, and a value greater than 0.7 means the observed variables are valid measurements for the latent variables. Average variance extracted (AVE) for all exogenous and endogenous construct denotes the convergent validity and the value of 0.5 suggests the model meets the minimum requirement.

## 5. Results

The PLS path model developed in this study is estimated with SmartPLS (SmartPLS GmbH. https://www.smartpls.com/downloads) and the result is summarized in this section. The adjusted $R^2$ value of the structural model is 0.25, indicating 25% of the variance of the burglary rates can be explained by the model.

Table 2 provides the goodness-of-fit statistics of CR, CCA, and AVE. These statistics all exceed the required minimum values of 0.65, 0.7, and 0.5, indicating that all the measurement models are reliable [42]. In summary, both the measurement model and the structural model shows adequate model fitness, indicating that the PLS path model result can be accepted.

**Table 2.** The result of the model fit index.

|     | CR    | CCA   | AVE   |
| --- | ----- | ----- | ----- |
| SED | 0.861 | 0.766 | 0.674 |

### 5.1. Measurement Model Results

Figure 3 displays the standardized path coefficients for the model. The factor loadings of the SED variables of LowRent, Worker, and LowEdu are larger than 0.77, suggesting that they are strong indicators of SED. In this model, we only use the variable of renters to measure mobility because of the data availability. Based on previous studies, a community with a great number of renters has a high degree of mobility.

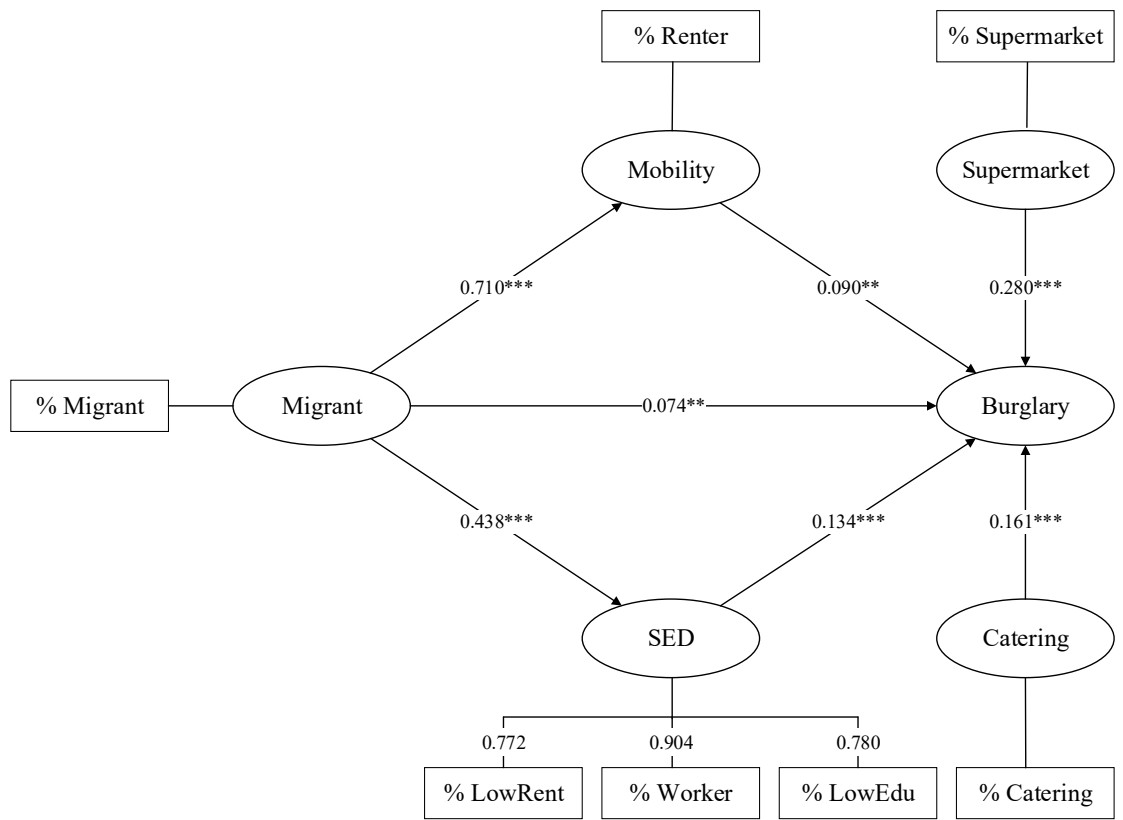

**Figure 3.** Standardized pathway coefficients (*** $p < 0.001$; ** $p < 0.05$).

### 5.2. Direct Effects

There are seven direct paths in the model, including five direct paths related to burglary rates and two direct links between migrants and residential mobility, and socio-economic disadvantage, respectively. As shown in Figure 3, all the path coefficients of migrants and other explanatory variables are statistically significant ($p < 0.05$).

The positive coefficient of 0.074 on the path from the percentage of migrants to burglary rates supports the first hypothesis on migrants directly contributing to burglaries. The paths between SED and burglary rates have the largest standard coefficient ($\beta = 0.134$) among all the direct paths to burglary rates, suggesting that SED is the most influential variable affecting burglary rates. Meanwhile, residential mobility also has significant direct effects on burglary ($\beta = 0.09$), suggesting a community with greater residential mobility would experience a higher burglary rate. These findings are well explained by social disorganization theory, which states that the socially disorganized communities

tend to have a low collective efficacy. This will consequently cause a lack of guardianship, and subsequently more opportunities for burglars are generated in such communities [10,42].

Besides the direct effects on burglary rates, migrants also have significant direct impacts on the mediating factors. The percentage of migrants have positive relationships with mobility ($\beta = 0.710$) and SED ($\beta = 0.438$). This means that migrants tend to concentrate in the community with greater residential mobility and a large number of SEDs.

As for the control variables, we found out that both the amount of catering facilities and supermarkets per 1000 families in the community have positive impacts on burglary rates, indicating that burglary rates tend to be higher in communities with more catering facilities and supermarkets, which is in line with our expectations.

*5.3. Indirect Effects*

The indirect effect of an exogenous variable on an endogenous variable through a mediating variable is obtained by summation of the products of the coefficients of each of the two indirect pathways. The Migrant-Mobility-Burglary pathway has a coefficient of 0.064, equivalent of $0.710 \times 0.090$, and the Migrant-SED-Burglary pathway has a coefficient of 0.059, equivalent of $0.438 \times 0.134$. The coefficient for the total indirect effects is 0.123, equivalent of $0.064 + 0.059$. Table 3 provides the direct (0.074), indirect (0.123) and total effects (0.197) of migrants on burglary rates. Specifically, migrants impose positive indirect impacts on burglary rates through increasing residential mobility and enhancing SED in the community.

**Table 3.** Standardized indirect effects of migrants on burglary rates.

| Mediating Factors | | Standardized Coefficients | |
|---|---|---|---|
| Mobility | | 0.064 | |
| SED | | 0.059 | |
| **Total** | | | |
| | **Direct Effect** | **Indirect Effects** | **Total Effects** |
| % Migrants→burglary rates | 0.074 | 0.123 | 0.197 |

The total indirect effect of 0.123 is almost twice as large as the direct effect of 0.074. This implies that the impact of migrants on burglaries is largely shaped through the mediating effects of community residential stability and socio-economic characteristics. In other words, a community with a high proportion of migrants is more likely to witness its social structure altered and social ties among residents weakened due to high mobility and mixed cultures. This inevitably increases opportunities for offenders to commit crimes. Meanwhile, the SED of the community with a high proportion of migrants is generally high, which may cause greater disorder in the community and in turn make it attractive to burglars by creating more criminal opportunities.

Combining both direct and indirect effects of migrants on burglaries together, the total effect of migrants on burglary rates is 0.197, which again strengthens the positive relationship between migrants and burglary rates. Although the direct effects of migrants on burglary rates is relatively smaller than that of residential mobility and SED, the total effect of migrant turns out to be the largest. This indicates that considering the indirect effect of migrants can better reveal the underlying processes of how migrants have changed crime rates in a community.

**6. Conclusions and Discussion**

Despite a longstanding interest in the relationship between immigration and crime, there has not been an agreed conclusion among prior studies. This indicates that the impacts of immigration on crime are context-contingent. Compared to western countries where international immigration is prominent, rural to urban migrants have imposed great social challenges in China. This paper

advances this topic by modeling the complex relationships between migrants and burglary rates in ZG city from the perspectives of social organization, opportunity structure and cultural approaches. The contributions of this paper are three-fold.

First, using SEM, this research reveals the direct, indirect, and total effects of migrants on burglary rates. Although our empirical findings support the crime generating view as migrants have positive impacts on burglary rates both directly and indirectly, it was found that the indirect effect is much larger than the direct effect, suggesting that migrants largely contribute to burglary rates through the mediating effects of community characteristics including residential mobility and SED [8].

Second, the complexity of migrant impacts on burglary rates is revealed in this study. The direct path and two indirect paths through residential mobility and SED are statistically significant, which suggests that migrants influence burglary rates through different processes. On one hand, the large number of migrants directly contribute to more crime opportunities in the community [41]. On the other hand, a socially disorganized community with weak collective efficacy attract more crime opportunities, resulting in indirect effects of migrants on burglaries [19,20]. Therefore, the impact of migrants on the burglary results are from both direct and indirect processes.

Third, this empirical study enriches the empirical evidence related to this topic in China. The proportion of renters as a substituting indicator of residential mobility is proven to be significant, and so does the percentage of families with low rental costs as a surrogate of SED [46].

Given the limitation on data availability, there are limitations that should be addressed in future research. First, some key variables typically used in western criminology and sociology studies, such as the number of people that moved in/out in the past years and the percentage of the population below the poverty line, are not available now in this research. The use of proxy variables in this research may reduce the explanatory power of the model. Second, differences may exist between the number of registered migrants and actual migrants in the city. Third, the cultural diversity of immigrants has not been explicitly considered in this research. Although the cultural diversity in China is not as much as that in Europe, America and other western countries, immigrants with different cultures might still have different impacts on the opportunity structure and social disorganization of the communities. With the consideration of the co-existence of immigrants with different origins, the immigrant-crime relationship has been found to be quite complex in Western countries [28]. However, this issue has not been empirically examined in China. Additionally, in the current work, the effect of opportunity structure, social disorganization and cultures are still mixed. To single out each aspect's contribution to crime, a longitudinal study should be conducted in the future. For that study, social-economic status, residential mobility, immigrants' origins, and their interactions should be explicitly considered.

In conclusion, this study reveals the complex relationships between migrants and burglaries, which provides useful insights into how migrants affect urban safety. The findings may facilitate policy-makers and law enforcement agencies to make appropriate policies to alleviate issues caused by migrants. For example, they can better target communities with high proportions of migrants to implement policies that promote social integration and stability to reduce crimes.

**Author Contributions:** Conceptualization, F.D., L.L, and C.J.; methodology, F.D., L.L. and C.J.; software, F.D. and D.L.; formal analysis, F.D. and L.L.; writing—original draft preparation, F.D., L.L., C.J., D.L., and M.L.; writing—review and editing, F.D., L.L., and C.J.; supervision, L.L.; project administration, L.L.; funding acquisition, L.L.

**Funding:** This research was supported under the National Key R&D Program of China (Nos. 2018YFB0505500, 2018YFB0505503), Key Program of National Natural Science Foundation of China (No. 41531178), Key Project of Science and Technology Program of Guangzhou City, China (No. 201804020016), Research Team Program of Natural Science Foundation of Guangdong Province, China (No. 2014A030312010), National Science Fund for Excellent Young Scholars (No. 41522104) and Science and Technology Program of Guangdong Province, China (No. 2015A020217003).

**Acknowledgments:** The authors would like to express deep gratitude to Fang Ren, a respectable and responsible scholar, who has provided us with valuable comments and helped us edit the paper.

**Conflicts of Interest:** The authors declare no conflict of interest.

**Note:** Access to crime data was granted by the police authorities on the condition that the real name of the city would not be mentioned in publications.

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
