# Peer review of "Discerning the Effects of Rural to Urban Migrants on Burglaries in ZG City with Structural Equation Modeling"

_sustainability, doi:10.3390/su11030561_

Round 1
Reviewer 1 Report
Resilience Capacity and Sustainable Development:
Adaptation to External Shocks
I read the article with great interest the paper entitled Resilience Capacity and Sustainable Development: Adaptation to External Shocks
It is very well written and presented.
Only two questions: I would recommend the authors to enrich more the theoretical framework (vulnerability and precarity ) and link it more to the analysis.
I recommend update the bibliography
Author Response
Point 1: I would recommend the authors to enrich more the theoretical framework (vulnerability and precarity) and link it more to the analysis. I recommend update the bibliography
Response 1: We thank the reviewer for agreeing to review our article. However, it seems that the reviewer had submitted the review report for another article, not ours. The title of our paper is “Discerning the effects of rural to urban migrants on burglaries in ZG city with structural equation modeling”.
Reviewer 2 Report
This paper is well written and I enjoyed reading it. I have some reservations that should be considered before publication.
I agree that the evidence is mixed regarding immigration and crime. I refer you to the following paper (not mine) which might help clarify some of the issues you find:
Ignatans, Dainis and Roebuck, Timothy (2018) Do more immigrants equal more crime? Drawing a bridge between first generation immigrant concentration and recorded crime rates. Crime, Security and Society, 1 (1). ISSN 2398-130X
I think there are additional factors at play (at least in Western Society) that require further consideration in light of your results.
Revision One:
There are limitations to each of the theories you suggest in regards immigration and crime. You need to consider limitations of opportunity structure, cultural approaches and social disorganisation. You need to consider the limitations of these in relation to your model. These should be discussed in literature and then expanded upon against revision 6 in your discussion.
Revision Two:
You need to consider the possibility of the interaction between culture and social disorganisation in more detail. The Ignatans study suggests that
(i) The background of the immigrant population is less important, and what appears to be key is whether there is a cultural similarity among the immigrant population within an area. So like cultures moving to like cultural areas.
(ii) The research also showed that areas where two or more cultures (other than that of the indigenous population) are prevalent, tend to be very high in crime.
(iii) It also found differences between first and second generation immigrants.
This might suggest there is an interaction between culture and social disorganisation that is difficult to separate out. A hypothesis here is if immigrants move to more homogenous areas (in terms of culture/ethnicity) this might have a bigger impact on social disorganisation and hence crime than if they move to an already multi-cultural area.
It is important to consider how this might influence the China situation in terms of domestic migration.
Revision Three:
Your understanding of crime generators and crime attractors needs refining. An attractor is a place offenders go to because of a known reputation for crime. A generator is one that lots of people go to, and as part of that greater opportunities for offending result. There is no prior motivation to commit offences at a crime generator.
See Newton, A. (2018). Macro level generators and crime: Parks, stadiums, and transit Stations; in Bruinsma, G. and Johnson, S (Eds). The Oxford Handbook of Environmental Criminology: Oxford University Press
I am not convinced that supermarkets increase crime as offer better hiding places. I would argue offenders may travel there as part of their routine, and by doing so increase their awareness activity of potential burglary opportunities near to these large supermarkets. The ‘hiding’ argument seems less plausible, especially for burglary, and I would criticise this explanation. It also might reduce capable guardianship if residents are out of home and at the supermarket (for work or shopping).
Revision 4
I again wonder about your direct and indirect effects. I think key might be that the immigration can have different influences on social disorganisation. When immigration reduces social disorganisation this may impact greater on crime, which may occur when immigrants are more culturally different.
Revision 5
You consider this from the perspective of the immigrant as the offender but it is possible the offender is the victim, especially second generation migrants in UK. How might this influence your literature, analysis, and interpretation of your findings? This warrants consideration and discussion
Revision 6
You need to elaborate on limitations of this paper. This section is rather thin. What future research direction can improve this study. I think a longitudinal exploration here would be very important to tease out changes in social disorganisation and culture over time
Author Response
Thanks for your comments. We have studies the valuable comments from you carefully, and tried our best to revise the manuscript. The point to point responds are liseted as following:
Point 1: This paper is well written and I enjoyed reading it. I have some reservations that should be considered before publication.
I agree that the evidence is mixed regarding immigration and crime. I refer you to the following paper (not mine) which might help clarify some of the issues you find:
Ignatans, Dainis and Roebuck, Timothy (2018) Do more immigrants equal more crime? Drawing a bridge between first generation immigrant concentration and recorded crime rates. Crime, Security and Society, 1 (1). ISSN 2398-130X
I think there are additional factors at play (at least in Western Society) that require further consideration in light of your results.
Response 1:
Thanks for your comments and suggestions. We thoroughly read the paper you recommended and made revisions accordingly (please see individual responses below for details of the revisions). The paper also gives us ideas for future work.
Point 2: There are limitations to each of the theories you suggest in regards immigration and crime. You need to consider limitations of opportunity structure, cultural approaches and social disorganisation. You need to consider the limitations of these in relation to your model. These should be discussed in literature and then expanded upon against revision 6 in your discussion.
Response 2: Many thanks for this comment. There are certain limitations for the theories of opportunity structure, cultural approaches and social disorganization. We built our research and improved upon previous work that employs similar theoretical framework (Feldmeyer, 2009).
According to the reviewer’s comment, revisions have been made in section 2.3 and section 6.
[Section 2.3: Limitations, such as the lack of explicitly considering the effect of culturally different migrants who live in the same community [28], do exist for these three theories. However, the cultural issue in China is not as pronounced as that in Europe or America, therefore we consider the immigrant as a whole in this study.] We plan take this issue into consideration in our future studies.
[Section 6: Third, the cultural diversity of immigrants has not been explicitly considered in this research. Although the cultural diversity in China is not as much as that in Europe, America and other western countries, the immigrants with different cultures might still have different impacts on the opportunity structure and social disorganization of the communities. With the consideration of co-existence of immigrants with different origins, the immigrant-crime relationship has found to be quite complex in Western countries [28]. However, this issue hasn’t been empirically examined in China.]
Reference:
Feldmeyer, B. Immigration and violence: the offsetting effects of immigrant concentration on Latino violence. Social Science Research, 2009, 38(3), 717-731.
[28] Ignatans, D. and T. Roebuck (2018). "Do more immigrants equal more crime? Drawing a bridge between first generation immigrant concentration and recorded crime rates." Crime, Security and Society 1 (1).
Point 3: You need to consider the possibility of the interaction between culture and social disorganisation in more detail. The Ignatans study suggests that
(i) The background of the immigrant population is less important, and what appears to be key is whether there is a cultural similarity among the immigrant population within an area. So like cultures moving to like cultural areas.
(ii) The research also showed that areas where two or more cultures (other than that of the indigenous population) are prevalent, tend to be very high in crime.
(iii) It also found differences between first and second generation immigrants.
This might suggest there is an interaction between culture and social disorganisation that is difficult to separate out. A hypothesis here is if immigrants move to more homogenous areas (in terms of culture/ethnicity) this might have a bigger impact on social disorganisation and hence crime than if they move to an already multi-cultural area.
It is important to consider how this might influence the China situation in terms of domestic migration.
Response 3: Many thanks for this suggestion. The paper written by Ignatans and Roebuck has been read carefully and thoroughly. We found it helpful to understand the immigrant-crime relationship with the specific mediation of cultural origins.
According to the reviewer’s suggestion, we have made some revisions in section 2.2, section 2.3 and section 6 to discuss the interaction between culture and crime. WE would like to emphasize two relevant points. First, most immigrants in China share a similar culture, and thus the culture issue in China is not as pronounced as that in Europe or America. Second, the two smallest administration levels in China are Juwei and Jiedao. Jiedao usually consists of more than three Juweis, making it much more heterogeneous than Juwei and less suitable for community analysis. Our analysis is carried out at the Juwei level while the data on the origins of immigrants can only be accessed at the Jiedao level. For such reasons, we need to re-design the framework at the Jiedao level for analyzing the effects of immigrants’ culture on crime, which would lend itself to another paper.
[Section 2.2: From a cultural perspective, Ignatans and Roebuck (2018) found that the areas with the most European and African immigrants have the lowest average crime rates in England and Wales. Their results also suggest that the cultural similarity between the migrant and indigenous population is a key determinant of whether immigrants increase or decrease crime.]
[Section 2.3: Limitations, such as the lack of explicitly considering the effect of culturally different migrants who live in the same community [28], do exist for these three theories. However, the cultural issue in China is not as pronounced as that in Europe or America, therefore we consider the immigrant as a whole in this study.]
[Section 6: Third, the cultural diversity of immigrants has not been explicitly considered in this research. Although the cultural diversity in China is not as much as that in Europe, America and other western countries, the immigrants with different cultures might still have different impacts on the opportunity structure and social disorganization of the communities. With the consideration of co-existence of immigrants with different origins, the immigrant-crime relationship has found to be quite complex in Western countries [28]. However, this issue hasn’t been empirically examined in China.]
Reference:
[28] Ignatans, D. and T. Roebuck (2018). "Do more immigrants equal more crime? Drawing a bridge between first generation immigrant concentration and recorded crime rates." Crime, Security and Society 1 (1).
Point 4: Your understanding of crime generators and crime attractors needs refining. An attractor is a place offenders go to because of a known reputation for crime. A generator is one that lots of people go to, and as part of that greater opportunities for offending result. There is no prior motivation to commit offences at a crime generator.
See Newton, A. (2018). Macro level generators and crime: Parks, stadiums, and transit Stations; in Bruinsma, G. and Johnson, S (Eds). The Oxford Handbook of Environmental Criminology: Oxford University Press
I am not convinced that supermarkets increase crime as offer better hiding places. I would argue offenders may travel there as part of their routine, and by doing so increase their awareness activity of potential burglary opportunities near to these large supermarkets. The ‘hiding’ argument seems less plausible, especially for burglary, and I would criticise this explanation. It also might reduce capable guardianship if residents are out of home and at the supermarket (for work or shopping).
Response 4: Thanks for this comment. We agree with you on that supermarket should be viewed as a crime-generator rather than a crime-attractor. Corresponding revisions have been made.
[Past studies have shown that a burglar is more likely to choose his/her target near urban facilities like supermarkets and catering facilities because these places usually act as crime generators [38, 39]. By providing food and entertainment, supermarket and catering facilities are key routine activity nodes for both the citizens and potential burglars, and thus constrain the spatial range of their movement [40]. As a result, the potential burglars will be more familiar with the residential areas in the vicinity than those far away. Therefore, we expect the higher burglary rates to occur at the places with the high average number of supermarkets and catering facilities.]
Reference:
[38] Newton, A. Macro level generators and crime: Parks, stadiums, and transit Stations. In Bruinsma, G. and Johnson, S (Eds). The Oxford Handbook of Environmental Criminology. Oxford University Press, 2018.
[39] Liu, L.; Jiang, C.; Zhou, S., Liu, K.; Du F. Impact of public bus system on spatial burglary patterns in a Chinese urban context. Applied Geography, 2017, 89, 142-149.
[40] Chen, J.; Liu, L.; Zhou, S.; Xiao, L.; Jiang, C. Spatial Variation Relationship between Floating Population and Residential Burglary: A Case Study from ZG, China. ISPRS International Journal of Geo-Information, 2017, 6 (8), 1-16.
Point 5: I again wonder about your direct and indirect effects. I think key might be that the immigration can have different influences on social disorganisation. When immigration reduces social disorganisation this may impact greater on crime, which may occur when immigrants are more culturally different.
Response 5: Thanks for this comment. The direct and indirect paths in the SEM model are used to discern and identify the specific mechanism of immigrant’s impact on burglary. In our result, we identified three mechanisms for the immigrant-burglary relationship, two are shown by the indirect path and one is shown by the direct path.
We agree on that in different cultural context immigration may have different effects on social disorganization. However, for our study area, the structural equation model suggests that the overall effect of immigration is to increase social disorganization in the community, as the coefficients from migrant to residential mobility and social-economic disadvantages are both significant and positive, 0.710 and 0.438 respectively. As the cultural diversity in China is not as much as that in the Western countries, we would like to make clear the overall effect in this paper first and then in the next paper delve into the complex relationship with cultural differences into consideration.
Point 6: You consider this from the perspective of the immigrant as the offender but it is possible the offender is the victim, especially second generation migrants in UK. How might this influence your literature, analysis, and interpretation of your findings? This warrants consideration and discussion.
Response 6: Thanks for this comment. Based on the work of testing social disorganization theory by Robert Sampson and his colleges (1989, 1997), the high residential mobility and low social-economic status can reduce the community efficacy, which can lead to a lack of informal control over the potential offenders and insufficient guardianship over the potential targets within the community. In our conceptual framework and analysis, the immigrant can be either the offender or the victim. For example, in section 5.2 Direct effects, we interpret the model results as “These findings are well explained by the social disorganization theory, which states that the socially disorganized communities tend to have a low collective efficacy. This will consequently cause a lack of guardianship, and subsequently more opportunities for burglars are generated in such communities”. In section 5.3 Indirect effects, we find and state the result as “The total indirect effect of 0.123 is almost twice as large as the direct effect of 0.074. This implies that the impact of migrants on burglaries is largely shaped through the mediating effects of community residential stability and socio-economic characteristics. In other words, a community with a high proportion of migrants are more likely to witness its social structure to be altered and social ties among residents are weakened due to high mobility and mixed cultures. This inevitably increases opportunities for offenders to commit a crime”.
According to the reviewer’s comment, we made some revisions in section 3 to make our point clearer. [Based on the previous studies and the special context of China, we evaluated these indirect effects of migrants on crime through two mediating variables: residential mobility and socio-economic disadvantage. Specifically, we expect that the high residential mobility and low social-economic status would reduce the community efficacy, which can lead to a lack of informal control over the potential offenders and insufficient guardianship over the potential targets within the community, thus generating much more opportunities for the occurrence of burglary [19,20].]
Reference:
[19] Sampson, R. J. and W. B. Groves (1989). "Community Structure and Crime: Testing Social-Disorganization Theory." The American Journal of Sociology 94 (4): 774-802.
[20] Sampson, R. J. and S. W. Raudenbush, et al. (1997). "Neighborhoods and Violent Crime: A Multilevel Study of Collective Efficacy." Science 277 (5328): 918 -924.
Point 7: You need to elaborate on limitations of this paper. This section is rather thin. What future research direction can improve this study? I think a longitudinal exploration here would be very important to tease out changes in social disorganisation and culture over time.
Response 7:
Thanks for this suggestion. According to your suggestion, the discussion in section 6 has been enriched.
[Given the limitation on data availability, there are limitations that should be addressed in the future research. First, some key variables typically used in western criminology and sociology studies, such as the number of people that moved in/out in the past years and the percentage of population below the poverty line, are not available now at this research. The use of the proxy variables in this research may reduce the explanatory power of the model. Second, difference may exist between the number of registered migrants and actual migrants in the city. Third, the cultural diversity of immigrants has not been explicitly considered in this research. Although the cultural diversity in China is not as much as that in Europe, America and other western countries, the immigrants with different cultures might still have different impacts on the opportunity structure and social disorganization of the communities. With the consideration of co-existence of immigrants with different origins, the immigrant-crime relationship has found to be quite complex in Western countries [28]. However, this issue hasn’t been empirically examined in China. Additionally, in the current work, the effect of opportunity structure, social disorganization and cultures are still mixed. To single out each aspect’s contribution to crime, a longitudinal study should be conducted in the future. For that study, the social-economic status, residential mobility, immigrants’ origins, and their interactions should be explicitly considered.]